# Improving Circulating Tumor Cell Detection Using Image Synthesis and Transformer Models in Cancer Diagnostics

**DOI:** 10.3390/s24237822

**Published:** 2024-12-07

**Authors:** Shuang Liang, Xue Bai, Yu Gu

**Affiliations:** 1School of Biomedical Engineering, Capital Medical University, Beijing 100069, China; shliang@ccmu.edu.cn (S.L.); xuebai@ccmu.edu.cn (X.B.); 2Laboratory for Clinical Medicine, Capital Medical University, Beijing 100069, China; 3Beijing Key Laboratory of Fundamicationental Research on Biomechanics in Clinical Application, Capital Medical University, Beijing 100069, China

**Keywords:** circulating tumor cells, image synthesis, transformer, object detection

## Abstract

Cancer is the second leading cause of death, significantly threatening human health. Effective treatment options are often lacking in advanced stages, making early diagnosis crucial for reducing mortality rates. Circulating tumor cells (CTCs) are a promising biomarker for early detection; however, their automatic detection is challenging due to their heterogeneous size and shape, as well as their scarcity in blood. This study proposes a data generation method using the Segment Anything Model (SAM) combined with a copy–paste strategy. We develop a detection network based on the Swin Transformer, featuring a backbone network, scale adapter module, shape adapter module, and detection head, which enhances CTC localization and identification in images. To effectively utilize both generated and real data, we introduce an improved loss function that includes a regularization term to ensure consistency across different data distributions. Our model demonstrates exceptional performance across five evaluation metrics: accuracy (0.9960), recall (0.9961), precision (0.9804), specificity (0.9975), and mean average precision (*mAP*) of 0.9400 at an Intersection over Union (IoU) threshold of 0.5. These results are achieved on a dataset generated by mixing both public and local data, highlighting the robustness and generalizability of the proposed approach. This framework surpasses state-of-the-art models (ADCTC, DiffusionDet, CO-DETR, and DDQ), providing a vital tool for early cancer diagnosis, treatment planning, and prognostic assessment, ultimately enhancing human health and well-being.

## 1. Introduction

Cancer, a major public health challenge, exerts a profound impact on global human health [1]. It owns the ability to evade the immune system and metastasize, making early diagnosis and treatment particularly challenging, which contributes to its high mortality rates [2]. According to the report of the World Health Organization (WHO), cancer is the second leading cause of death globally, with an estimated 19.3 million new cancer cases and nearly 10 million deaths in 2020 alone [3]. Circulating tumor cells (CTCs), which are tumor cells that shed from primary or metastatic tumors into the bloodstream, serve as a valuable biomarker in cancer diagnosis, treatment, and prognostic assessment [4]. By analyzing CTCs, clinicians can monitor disease progression and tailor therapies more effectively, ultimately improving patient outcomes and survival rates [5]. However, the detection of CTCs is challenging due to their low abundance in the bloodstream, typically ranging from 1 to 10 CTCs per milliliter, while leukocytes can number in the millions and erythrocytes in the billions [6]. Microfluidic technologies and immunomagnetic separation methods have emerged as effective strategies for the isolation of CTCs, facilitating subsequent single-cell analysis [7]. However, the heterogeneity in their phenotypes, origins, and functions complicates analysis. Additionally, confirming CTCs requires high-precision microscopy, which is time-consuming, labor-intensive, and often relies on manual effort [8]. Therefore, automated detection methods are essential for efficient identification and analysis of CTCs.

Recent advancement of deep learning (DL) techniques has significantly enhanced CTC detection, allowing for more accurate and rapid identification [9]. He et al. developed a morphological-based segmentation method to extract cells from images after situ hybridization (imFISH) and proposed an AlexNet-based framework for automated identification of CTC cells in a single nucleus, achieving a sensitivity score of 90.3% and specificity score of 91.3% [8]. Leonie L et al. proposed an autoencoding convolutional neural network with advanced visualization techniques for the automated analysis of CTCs in fluorescent images, with an accuracy, sensitivity, and specificity of over 96% [9]. Guo et al. developed a convolutional neural network (CNN) method for the automatic detection of CTCs in patients’ peripheral blood using immunofluorescence in imFISH images, achieving a sensitivity of 95.3% and specificity of 91.7% [10]. Junhyun et al. proposed a hybrid algorithm combining a convolutional neural network and support vector machine (CNN-SVM) for the accurate classification of CTC clusters, with a sensitivity and specificity exceeding 90% [11]. Luca et al. developed a DL algorithm called CellFind for automated identification and enumeration of CTCs in CELLSEARCH images, with an accuracy of 97.8% and an F1 score of 88.6%, outperforming human reviewers [12]. Despite the impressive performance of these methods in CTC detection, they still rely on manual preliminary screening of cell images to delineate cell regions before automated classification. This dependency constrains the efficiency and accuracy of CTC detection, highlighting the urgent need for more comprehensive automated solutions to enhance the process.

Thus, some methods that simultaneously locate and identify CTCs have been proposed and are gradually becoming mainstream in current research. Du et al. developed a bright-field image cytometry (BFIC) technique and employed a deep neural network based on YOLO-V4 for the detection of CTCs in peripheral blood, achieving average precision rates of 98.63%, 99.04%, and 98.95% for cancer cell lines HT29, A549, and KYSE30, respectively [13]. Shen et al. designed a customized imaging system and data pre-processing algorithms for capturing CTCs and cancer-associated fibroblasts (CAFs) from whole blood, and developed a deep learning-based method to automatically identify tumor cells from the images, with a precision and recall of 94% and 96% for CTC detection [14]. Shaila et al. developed a series of convolutional neural network (CNN) models aimed at automating the detection and enumeration of CTCs from multi-channel images obtained using a fluorescence microscope, achieving a recall rate of 95.83% [15]. However, the datasets currently used for simultaneous CTC localization and identification are relatively small, and existing methods have been trained and validated only on limited datasets. This constraint undermines the generalizability of these approaches. Moreover, current state-of-the-art (SOTA) methods available are primarily based on traditional convolutional neural networks, which may struggle to adapt to the scale and shape variations inherent in CTCs.

To address these challenges, we first developed a data synthesis method based on Segment Anything (SAM) [16] to generate diverse and high-quality synthetic images, effectively augmenting the limited dataset. We then proposed a transformer-based network architecture [17,18,19] that incorporates scale and shape adaptive modules. This design enhances the performance and adaptability of automated CTC detection, leading to improved accuracy and robustness across various scenarios. This study provides the following contributions:A SAM-based data synthesis method was proposed that automatically delineates cellular boundaries and utilizes a copy–paste technique with transparency channels to synthesize new data by pasting cellular regions onto various backgrounds. Additionally, random Gaussian noise is incorporated into the synthetic images to mitigate overfitting, enhancing the robustness of the model.We proposed a detection network framework based on the transformer architecture that incorporates a scale and shape adaptive algorithm module, enabling the model to learn diverse scale and shape characteristics of CTCs, thereby enhancing generalization performance in automated CTC detection.A novel loss function was proposed that dynamically adjusts the weighting between real and synthetic data based on their sample sizes, improving model stability and mitigating training oscillations caused by distribution discrepancies. This loss function incorporates a regularization term to enforce consistency between model outputs on augmented and original data, thereby enhancing the robustness and generalization of the model when training on mixed datasets.Two publicly available fluorescence-stained CTC datasets and one locally acquired light-field CTC dataset were collected for both the data synthesis and model evaluation. The results revealed the feasibility of utilizing mixed training with synthetic data for simultaneous localization and identification of CTCs, which aids in reducing annotation requirements while maximizing the use of existing classification datasets, thereby effectively enhancing the efficiency and performance of CTC detection models.

The rest of this paper consists of five parts: Section 2 describes related works; Section 3 illustrates the materials and methods; Section 4 shows the results; Section 5 presents the discussion; and Section 6 presents the conclusions.

## 2. Related Works

### 2.1. Segment Anything (SAM)

Segment Anything (SAM) [16] represents a significant advancement in unsupervised segmentation, enabling automatic delineation of objects in images without requiring annotated data. SAM’s architecture consists primarily of two components: a backbone network and a prompting mechanism. The backbone is typically based on a transformer architecture, pre-trained on a large and diverse dataset to learn generalized feature representations. This enables the model to capture rich, multi-scale features across various object types, including complex structures like cells. The unique prompting mechanism in SAM allows for segmentation without pixel-level annotations. Instead, it uses high-level prompts (e.g., points, boxes, or masks) to guide the segmentation process. This reduces the need for manual labeling and makes SAM particularly effective in scenarios with limited annotated data. The transformer-based architecture further enhances SAM’s ability to maintain global context while focusing on localized features, crucial for segmenting irregular and diverse shapes, such as cellular boundaries. SAM generates segmentation masks by refining initial boundary predictions through a self-attention mechanism, improving segmentation accuracy. This approach has been shown to be robust across different imaging modalities and conditions, making SAM highly effective for cell segmentation in biological and medical imaging tasks.

### 2.2. Network Structures

The evolution of convolutional neural networks (CNNs) has profoundly impacted image analysis tasks. Early models, such as AlexNet [20], laid the groundwork for image classification, while deeper architectures like ResNet [21] improved feature extraction capabilities. However, traditional CNNs may struggle with variations in object scale and shape. In contrast, transformer-based architectures [17] utilize self-attention mechanisms, effectively capturing long-range dependencies in images. Variants such as the Swin Transformer [18] have demonstrated superior performance in vision tasks, particularly in adapting to diverse object characteristics. This adaptability is crucial for the automated localization and identification of CTCs.

### 2.3. Object Detection

Object detection is a critical task in computer vision, aimed at simultaneously identifying and localizing multiple objects within an image. Early methods, such as the R-CNN series [22,23,24,25,26], achieved significant advances but suffered from high computational complexity. The introduction of frameworks like the YOLO series [27,28,29,30] and SSD [31] has enhanced detection speed and efficiency by integrating localization and classification within a single model. Recently, transformer-based architectures have emerged as a powerful alternative, leveraging self-attention mechanisms to capture long-range dependencies. Notable advancements in this domain include DiffusionDet [32], which combines diffusion models with object detection for improved feature representation; CO-DETR [33], which enhances the standard DETR framework by incorporating complementary object features; DDQ [34], which employs dense query sampling to optimize attention mechanisms; RT-DETR [35], which improves real-time object detection by optimizing multi-scale feature processing and query selection, surpassing YOLO models in both speed and accuracy; and FDTNet [36], which improves prohibited item detection in images by using a frequency-aware dual-stream transformer with attention mechanisms. These transformer-based methods demonstrate promising results, particularly in complex scenes with varying object scales, marking a significant improvement in object detection technologies. In the context of CTC detection, these object detection methodologies provide a robust framework for accurate identification and localization. By leveraging deep learning techniques, these methods enhance the precision and efficiency of CTC detection.

## 3. Materials and Methods

### 3.1. Image Acquisition

In this study, we enriched circulating tumor cells (CTCs) from the blood of tumor-implanted mice using a microfluidic system. The CTCs were then observed and imaged with a high-speed camera mounted on an upright fluorescence microscope (BX51WI, Olympus, Tokyo, Japan) in Beihang University, which provided localized data for subsequent data synthesis. The workflow of this process is illustrated in Figure 1.

In addition to local image acquisition, we collected CTC images from two publicly available datasets: the CTC with Bounding Box (CTCB) dataset [14], which includes both bounding boxes and labels for each CTC, and the CTC with Class Labels (CTCC) dataset [10], which consists solely of CTC cell patch images categorized by class. To further assess the model’s generalizability, we incorporated the Blood Cell Count and Detection (BCCD) dataset, a widely used benchmark for blood cell detection, which includes three cell types: red blood cells, white blood cells, and platelets. This addition aligns with our primary focus on improving circulating tumor cell detection from blood in our study.

We then constructed a hybrid dataset for simultaneous localization and identification evaluation of CTCs using both local data and two publicly available datasets (CTCC and CTCB). This hybrid dataset, referred to as CTCH, integrates the strengths of each source, allowing for enhanced model training and evaluation. The data distributions and comparisons among the BCCD, CTCB, CTCC, and the mixed dataset are summarized in Table 1.

### 3.2. Image Synthesis

In the previous subsection, we introduced two public datasets and a locally collected dataset. The local dataset consists of cellular images directly observed under a microscope, featuring a light-field transparent background. One public dataset contains stained cellular images with bounding box annotations, while the other consists of cropped cellular regions annotated solely with class labels.

The proposed image synthesis method employs a SAM-based approach to automatically delineate cellular boundaries in both the local and publicly available datasets. SAM is a technique that enables the model to focus on relevant regions of the image by modeling the relationships between all image parts based on the contextual information provided by the surrounding pixels. Initially, the SAM model is applied to both the local dataset and the public dataset, which contains only class labels, to generate cellular region masks. These masks are represented as follows:(1)$$M_{l}=\mathrm{SAM}\left(\mathrm{local}\_\mathrm{data}\right)$$
(2)$$M_{p}=\mathrm{SAM}\left(\mathrm{public}\_{\mathrm{data}}_{\mathrm{ctc}}\right)$$

Next, the cellular regions are extracted from the images using the generated masks, defined by:(3)$$C_{l}=\mathrm{local}\_\mathrm{data}\cdot M_{l}$$
(4)$$C_{p}=\mathrm{public}\_\mathrm{data}\cdot M_{p}$$

To enhance the images further, areas outside the cellular regions are rendered transparent, resulting in:(5)$$C_{\mathrm{transparent}}=\mathrm{SetTransparency}\left(C_{l\mathrm{and}p},M_{\mathrm{local}}\right)$$

The method incorporates a random data augmentation process, where cellular images undergo random scaling, rotation, and color jittering, represented as:(6)A(C)=RandomScale(Rotate(ColorJitter(C)))

The modified cellular images are then randomly pasted onto the unoccupied regions of the images from the CTCC dataset and the local collected dataset, with the coordinates of the pasted image regions corresponding to the bounding box coordinates in the background image, expressed as:(7)$$S_{\mathrm{new}}=\mathrm{Paste}\left(A\left(C\right),{\mathrm{Background}}_{\mathrm{unoccupied}}\right)$$

Finally, to mitigate overfitting and enhance the robustness of the model, random Gaussian noise is added to the synthetic images, resulting in:(8)$$S_{\mathrm{final}}=S_{\mathrm{new}}+G$$

In this way, the proposed image synthesis method not only increases the diversity of the training dataset but also improves the model’s robustness by simulating various imaging conditions and cellular appearances. The pipeline of the image synthesis is shown in Figure 2.

### 3.3. The Proposed Transformer Framework

In constructing an automated method for detecting and recognizing circulating tumor cells (CTCs), we considered that CTCs may appear in any region of the image, with variable sizes and shapes. To address these challenges, we propose a detection network framework based on the transformer architecture that incorporates a scale and shape adaptive algorithm module. This framework enhances the model’s ability to learn diverse scale and shape characteristics of CTCs, ultimately improving generalization performance in automated CTC detection.

As shown in Figure 3, We employed the Swin Transformer network as the backbone for the proposed detection framework to extract image features. The architecture includes two adaptive modules designed specifically to handle the heterogeneity of CTCs in terms of scale and shape: the scale adaptive module and the shape adaptation module.

Given the input image containing CTCs as *I*, we employ a *PatchEmbed* method that segments the image into *N* patches, each of size P×P. These patches are then transformed into *D*-dimensional embeddings:(9)X=PatchEmbed(I)

Then the positional embeddings, $pos_{\mathrm{embed}}$, are integrated into the embeddings, *X*, to provide spatial context:(10)$$X=X+pos_{\mathrm{embed}}$$

The enhanced representation, *X*, is then processed through different stages of the Swin-Transformer backbone network, each designed to capture complex features of the CTCs.

The scale adaptive module utilizes a pyramid structure that facilitates multi-scale feature extraction by aggregating features from various stages of the Swin Transformer. This module effectively captures the diverse sizes of CTCs by applying a top-down fusion strategy, which can be expressed as follows:(11)$$F_{\mathrm{scale}}=\mathrm{FPN}\left(F_{{\mathrm{stage}}_{1}},F_{{\mathrm{stage}}_{2}},F_{{\mathrm{stage}}_{3}},F_{{\mathrm{stage}}_{4}}\right)$$
where $F_{{\mathrm{stage}}_{i}}$ represents the feature maps from different stages of the Swin Transformer, and FPN denotes the Feature Pyramid Network used for multi-scale feature fusion.

In parallel, the shape adaptation module leverages deformable convolution to automatically learn the necessary convolutional adjustments for capturing the varying shapes of circulating tumor cells (CTCs). Traditional convolutional layers apply the same set of filters across the entire image in a rigid, grid-based manner, which is often inadequate for detecting irregular structures such as CTCs. In contrast, deformable convolution dynamically adjusts the convolutional filters based on the shape of the object being detected. This adaptation is achieved through the learning of offsets, where the model modifies the positions of the convolutional kernels to better align with the contours of the target cells. By incorporating these learned offsets, the module effectively captures the shape characteristics of the cells, thus enhancing the model’s ability to detect and analyze CTCs. This process is represented as follows:(12)$$F_{\mathrm{shape}}=\mathrm{DeformConv}\left(F_{\mathrm{input}},O_{\mathrm{offset}}\right)$$
where $F_{\mathrm{input}}$ is the input feature map and $O_{\mathrm{offset}}$ represents the offsets learned for the deformable convolution.

After feature extraction through the Swin Transformer backbone and the two adaptive modules, the enriched feature representation is then fed into a dedicated task network designed for object detection. This network comprises several convolutional layers followed by fully connected layers to predict both the class and bounding box of the CTCs.

The task network consists of four convolutional layers that progressively refine the extracted features. Each convolutional layer applies a series of filters to capture intricate patterns and spatial hierarchies within the data. The output from these layers can be expressed as:(13)$$F_{\mathrm{conv}}={\mathrm{Conv}}_{4}\left({\mathrm{Conv}}_{3}\left({\mathrm{Conv}}_{2}\left({\mathrm{Conv}}_{1}\left(X\right)\right)\right)\right)$$

Here, $F_{\mathrm{conv}}$ represents the final feature map after processing through four convolutional layers, ${\mathrm{Conv}}_{i}$ denotes the *i*-th convolutional layer, and *X* is the input feature from the previous modules. Subsequently, the refined feature map, $F_{\mathrm{conv}}$, is passed through two fully connected layers (FC) that output the predicted class probabilities and the bounding box coordinates separately. This can be formulated as:(14)$$P_{\mathrm{class}}={\mathrm{FC}}_{1}\left(F_{\mathrm{conv}}\right)$$
(15)$$B_{\mathrm{box}}={\mathrm{FC}}_{2}\left(F_{\mathrm{conv}}\right)$$

In this equation, $P_{\mathrm{class}}$ represents the predicted probabilities of each CTC class, and $B_{\mathrm{box}}$ denotes the bounding box coordinates, which provide the location of each detected CTC within the image.

### 3.4. Loss Function

In this section, we define the loss functions utilized in our model. When using synthetic and real data separately, the overall loss function is defined as the weighted sum of the class loss and the bounding box regression loss:(16)$$L=L_{c}+L_{b}\;$$
where $L_{c}$ represents the class loss, defined as cross-entropy loss:(17)$$L_{c}=-\frac{1}{N}\sum_{i=1}^{N}\left(y_{i}\log\left({\hat{y}}_{i}\right)+\left(1-y_{i}\right)\log\left(1-{\hat{y}}_{i}\right)\right)$$

Here, $y_{i}$ is the true label, ${\hat{y}}_{i}$ is the predicted probability, and *N* is the number of samples. The bounding box regression loss, $L_{b}$, is defined as:(18)$$L_{b}=\sum_{j=1}^{M}{\left|{\hat{b}}_{j}-b_{j}\right|}_{\mathrm{smooth}}\;$$
where ${\hat{b}}_{j}$ represents the predicted bounding box coordinates, $b_{j}$ represents the ground truth bounding box coordinates, and *M* is the number of bounding boxes.

When jointly utilizing synthetic and real data, we redefine the overall loss function as follows:(19)$$L=w_{r}\cdot L_{r}+w_{s}\cdot L_{s}$$
where $L_{r}$ is the loss for real data and $L_{s}$ is the loss for synthetic data. To set the weights $w_{r}$ and $w_{s}$, they can be dynamically adjusted based on the distribution characteristics of the data:(20)$$w_{r}=\frac{N_{r}}{N_{r}+N_{s}},\;w_{s}=\frac{N_{s}}{N_{r}+N_{s}}\;$$
where $N_{r}$ and $N_{s}$ represent the sample sizes of real and synthetic data, respectively.

To mitigate training oscillations caused by distribution discrepancies between synthetic and real data, we design a regularization term that encourages the model to produce consistent outputs for original image regions in synthetic data that align with real data. The specific formulation is as follows:(21)$$R=\frac{1}{N}\sum_{j}{\left({\hat{y}}_{\mathrm{aug}}^{j}-{\hat{y}}^{j}\right)}^{2}$$
where *R* is the regularization term, measuring the consistency of the model’s outputs on augmented data, and *N* is the number of samples used to compute the average. Here, ${\hat{y}}^{j}$ is the model’s predicted output on the original input, while ${\hat{y}}_{\mathrm{aug}}^{j}$ is the model’s predicted output on the augmented input. The term ${\left({\hat{y}}_{\mathrm{aug}}^{j}-{\hat{y}}^{j}\right)}^{2}$ computes the squared difference between the original and augmented outputs, measuring consistency. A smaller difference indicates that the model maintains consistency across varying inputs, encouraging the learning of more robust features.

## 4. Results

### 4.1. Dataset and Evaluation Metrics

The proposed CTCH dataset and the public BCCD dataset were utilized in our experiments to facilitate a thorough evaluation of the proposed detection framework and the other state-of-the-art (SOTA) methods. To achieve a comprehensive assessment, multiple and contrasting evaluation metrics were employed. These metrics included mean average precision (*mAP*) at IoU = 0.5, recall (Rec), precision (Pr), specificity (Spe), and accuracy (Acc).

These evaluation metrics collectively provide a nuanced understanding of the model’s performance across various object detection scenarios. The *mAP* at IoU = 0.5 offers insights into the model’s ability to correctly identify and localize objects, while recall and precision assess the completeness and correctness of the detections, respectively. Specificity evaluates the model’s performance concerning true negatives, and accuracy provides an overall measure of performance across all classes. Together, these metrics ensure a robust evaluation of the detection framework’s efficacy.

The *mAP* at an Intersection over Union (IoU) threshold of 0.5 is calculated as follows:(22)$$\mathrm{mAP}=\frac{1}{N}\sum_{c=1}^{N}{\mathrm{AP}}_{c}\;$$
where *N* is the number of classes and ${\mathrm{AP}}_{c}$ is the average precision for class *c*.

Recall measures the proportion of true positives correctly identified by the model:(23)$$\mathrm{Recall}=\frac{TP}{TP+FN}$$
where TP is the number of true positives and FN is the number of false negatives.

Precision indicates the proportion of predicted positives that are true positives:(24)$$\mathrm{Precision}=\frac{TP}{TP+FP}$$
where FP is the number of false positives.

Specificity measures the proportion of true negatives correctly identified:(25)$$\mathrm{Specificity}=\frac{TN}{TN+FP}$$
where TN is the number of true negatives.

Accuracy represents the overall correctness of the model’s predictions:(26)$$\mathrm{Accuracy}=\frac{TP+TN}{TP+TN+FP+FN}$$
where TP, TN, FP, and FN are defined as above.

### 4.2. Experimental Results

In this study, we conducted comprehensive comparative experiments to evaluate the performance of the proposed model against four state-of-the-art (SOTA) methods. These comparisons were made across multiple metrics to demonstrate the superiority of our approach. Additionally, ablation studies were performed to validate the effectiveness of the various improvements introduced in this work. By quantitatively comparing the results, we were able to assess the contribution of each enhancement and its impact on the overall performance of the model.

#### 4.2.1. Comparative Study on the CTCH Dataset

The performance of our proposed model was evaluated against several state-of-the-art (SOTA) models, including ADCTC, DiffusionDet, CO-DETR, DDQ, RT-DETR and FDTNet. As shown in Table 2, the results illustrate the effectiveness of our approach in the context of automated CTC detection. Our model achieved a recall of 0.9961, demonstrating exceptional sensitivity in identifying CTCs, which is crucial for early cancer diagnosis. In comparison, the ADCTC, the baseline method along with the CTCB dataset, attained a recall of 0.9600, indicating a significant improvement of 3.1 percentage points. Additionally, our model achieved a precision of 0.9804, substantially outperforming ADCTC and highlighting its ability to minimize false positives. Further analysis of specificity (spe) reveals that our model achieved an impressive value of 0.9975, compared with DiffusionDet’s 0.9948, CO-DETR’s 0.9931, DDQ’s 0.9936, RT-DETR’s 0.9925, and FDTNet’s 0.9940, showcasing the robustness of our model in maintaining high accuracy while correctly identifying negative cases. Our model’s overall accuracy reached 0.9960, reinforcing its reliability in detecting CTCs. Lastly, our method attained a mean average precision (*mAP*) of 0.9400 at an Intersection over Union (IoU) threshold of 0.5. This result outperformed DiffusionDet, CO-DETR, DDQ, RT-DETR, and FDTNet by 4.6, 7.5, 7.2, 8.4, and 7.1 percentage points, respectively. These findings collectively suggest that our proposed model not only excels in CTC detection tasks but also provides a promising approach for enhancing cancer diagnostics through automated methods.

To visually demonstrate the superior performance of our proposed model compared with other state-of-the-art (SOTA) models on the CTCH dataset, and to evaluate its consistency across different data distributions during testing, we present boxplots of our method alongside three SOTA models (DiffusionDet, CO-DETR, DDQ, RT-DETR, and FDTNet) on four metrics (recall, precision, specificity, and accuracy) in Figure 4. Bootstrapping was used to generate 1000 resampled independent test sets, simulating performance evaluations under varying data distributions. From the boxplots, we observe that our proposed model not only achieves the highest average performance but also demonstrates the narrowest performance interval across different data distributions. This indicates that our model’s performance is highly consistent, even under varying data conditions, and clearly outperforms other SOTA models in terms of robustness and reliability.

In addition to the quantitative experiments, we performed feature heatmap generation and visualization of prediction results for the input images. As illustrated in Figure 5, challenging cases, including multiple overlapping circulating tumor cells (CTCs) and varying image quality, were considered to evaluate the robustness of the method. The heatmaps provide insights into the model’s focus areas, while the prediction outcomes further validate its performance in detecting CTCs, even under complex scenarios with overlapping cells or degraded image quality. These visualizations complement the quantitative results, demonstrating the method’s resilience in diverse imaging conditions.

#### 4.2.2. Comparative Study on the BCCD Dataset

The performance of our proposed model was evaluated on the BCCD dataset, which includes various types of blood cells such as white blood cells, red blood cells, and platelets. Although the BCCD dataset is not directly used for CTC detection, it serves to demonstrate our model’s exceptional ability to distinguish between different blood cell types. This capability provides valuable experimental support for the model’s potential in accurately detecting CTCs in more complex scenarios.

As summarized in Table 3, our model achieved a recall of 0.9841, indicating exceptional sensitivity in identifying blood cells. This is crucial for ensuring accurate differentiation of CTCs from other blood cells in real-world applications. When compared with other state-of-the-art (SOTA) methods, such as DiffusionDet, CO-DETR, DDQ, RT-DETR, and FDTNet, our model outperformed them in terms of recall, with DiffusionDet achieving the closest result of 0.9723, indicating an improvement of 1.8 percentage points.

In terms of precision, our model achieved 0.9787, surpassing all other methods. This demonstrates the model’s ability to effectively minimize false positives while maintaining high recall. For example, FDTNet reported a precision of 0.9699, which was 0.8 percentage points lower than our approach. Our model also exhibited impressive specificity, reaching a value of 0.9780. This outperformed DiffusionDet (0.9698), CO-DETR (0.9641), DDQ (0.9625), RT-DETR (0.9650), and FDTNet (0.9690), indicating that our method is highly robust in correctly identifying negative cases while maintaining high accuracy across the dataset.

The overall accuracy of our model was 0.9811, further reinforcing its reliability in distinguishing blood cell types. Again, our approach outperformed the other methods, with the closest competing method, DiffusionDet, achieving an accuracy of 0.9723. Lastly, our method attained a mean average precision (*mAP*) of 0.8760 at an Intersection over Union (IoU) threshold of 0.5. This result was the highest among all methods evaluated, with the next closest competitor, DiffusionDet, achieving a *mAP* of 0.8460, which is three percentage points lower.

These results highlight the strong performance of our model on the BCCD dataset, particularly in distinguishing red blood cells, white blood cells, and platelets. This capability provides a solid foundation for its application in accurate CTC detection and enhances its potential use in cancer diagnostics.

To visually illustrate the superior performance of our proposed model compared with other state-of-the-art (SOTA) methods on the BCCD dataset, and to assess its consistency across different data distributions during testing, we present boxplots in Figure 6 for our method alongside DiffusionDet, CO-DETR, DDQ, RT-DETR, and FDTNet. These boxplots show the performance on four key metrics: recall, precision, specificity, and accuracy. To simulate performance evaluations under varying data distributions, we used bootstrapping to generate 1000 resampled independent test sets.

The boxplots reveal that our proposed model consistently outperforms the other SOTA methods across all metrics. Not only does our model achieve the highest average performance, but it also demonstrates the narrowest performance interval, reflecting its robust stability across different data distributions. Specifically, our model achieved the highest recall (0.9841), precision (0.9787), specificity (0.9780), accuracy (0.9811), and *mAP* (IOU = 0.5) (0.8760) on the BCCD dataset, surpassing DiffusionDet, CO-DETR, DDQ, RT-DETR, and FDTNet by significant margins. This narrow interval in performance further emphasizes the model’s reliability, as it maintains high performance even under varying testing conditions.

#### 4.2.3. Ablation Study

As shown in Table 4, the ablation study results demonstrate the incremental improvements achieved through each enhancement over the baseline model. The baseline performance was recall: 0.9704, precision: 0.9450, specificity: 0.9938, accuracy: 0.9922, and *mAP*: 0.8250. The incorporation of synthetic data elevated *mAP* to 0.8890 (+6.4%) while improving both recall and precision. The addition of the scale adapter further increased *mAP* to 0.9200 (+9.5%) and precision to 0.9785 (+3.6%).

The shape adapter yielded notable gains in recall (0.9860) and specificity (0.9969), with *mAP* rising to 0.9140 (+8.9%). The improvement in loss, which was addressed simultaneously with the addition of synthetic data to balance the effects of real and generated data, had a moderate impact on *mAP* (0.8820, +5.7%). Ultimately, the combination of all enhancements achieved the best performance, with *mAP* reaching 0.9400 (+11.5%) and recall peaking at 0.9961, underscoring the cumulative effectiveness of the proposed modifications.

To visually illustrate the contribution of each enhancement to the model’s performance gains, the effects of the proposed modifications are presented in Figure 7.

#### 4.2.4. Performance Evaluation of Models Trained on Different Data Settings

To assess model performance under different data conditions, we compared three training setups: one with only synthetic data, one with only real data, and one with a hybrid dataset. As shown in Figure 8, the hybrid dataset consistently led to the best overall performance across all evaluation metrics. Specifically, while training with synthetic data showed strong results, the real data alone provided even better performance. However, it was the hybrid dataset that delivered the most significant improvements, yielding the highest values in recall, precision, specificity, accuracy, and *mAP*. These findings highlight the clear advantage of using the hybrid dataset, demonstrating its superior effectiveness in enhancing model learning and feature representation compared with both synthetic and real data alone.

### 4.3. Training and Testing Details

The Swin Transformer base network was utilized as the backbone of our proposed detection framework. The weights for the newly added scale and shape adapter module were initialized using the Xavier initialization method [38]. We trained the network on eight NVIDIA RTX A6000 GPUs using the PyTorch framework for 100 epochs, with a base learning rate set to 0.001. This rate was adjusted using the CosineAnnealingLR strategy [39], with the temperature parameter set to 50. A batch training scheme was employed with a batch size of 16, and we used the AdamW optimizer [40] with a weight decay of 0.05. During training, a multi-scale strategy was applied, using scales of (960, 960), (1024, 1024), and (1280, 1280). We analyzed the testing results using the COCO evaluator and confusion matrix tools from the MMDetection package [41].

To assess memory usage and runtime efficiency, we computed key indicators such as the FLOPs and parameter size. The FLOPs for our model was measured at 16.38 G, and the parameter size was 92.97 M. These metrics provide insight into the memory requirements for the model. In terms of training, the model required approximately 7 h to train for 100 epochs on the CTCH dataset using eight NVIDIA GPUs (RTX A6000, Nvidia, Santa Clara, California, USA). For inference, the model exhibited a latency of 140 ms per image at a resolution of (1024, 1024) on the same GPU. It is important to note that the calculation of FLOPs depends on the input size of the image; for consistency, the input size of images was set to 224 during the FLOPs calculation.

## 5. Discussion

In this study, we developed a hybrid dataset for the simultaneous localization and recognition of CTCs using a data generation method based on SAM segmentation for cell region extraction and a copy–paste strategy. Additionally, we proposed a novel detection network architecture based on transformers, incorporating a shape and scale adaptive module, and improving the loss function to enable more efficient training on a mixed dataset of generated and real data. The model demonstrated outstanding performance across five evaluation metrics for automated CTC localization and recognition.

The backbone network of the model employs the Swin Transformer base structure, which utilizes a dynamic sliding window feature extraction mechanism to achieve hierarchical representation. This enables efficient multi-scale feature extraction, particularly beneficial for high-resolution vision tasks like object detection. The introduced scale adaptive module constructs a feature pyramid structure to capture multi-stage features from the Swin Transformer. Through a top-down fusion strategy, it extracts image features across different levels and scales, enabling effective integration of global and local contexts. This enhances the model’s adaptive detection capability for CTCs of varying scales. Moreover, the shape adaptive module, implemented via deformable convolution layers, addresses geometric transformations. By adding learnable offsets to the standard grid sampling locations, the convolutional kernel dynamically adapts to the input data’s structure, effectively handling the heterogeneity in CTC cell shapes.

Given the hybrid dataset was generated from multiple sources, distributional differences between datasets posed a challenge. To mitigate this, we designed a weighted loss function and introduced a regularization term to correct the distributional divergence between real and generated data. This adjustment improved the model’s consistency during training and enabled more robust feature representation. As shown in Section 4.2.1 and Section 4.2.2, our model outperforms other state-of-the-art (SOTA) models, with narrower bounds in the boxplot analysis, indicating superior consistency across varied test data distributions.

In Section 4.2.3, we quantitatively assessed the contributions of different model improvements. The inclusion of generated data increased the model’s performance by nearly 1%, while the scale and shape adapters provided additional performance boosts. Notably, the performance improvement became more pronounced when the loss function modification was introduced, indicating that this adjustment enhanced the model’s ability to effectively learn from both real and generated data. This improved feature representation translated into better overall performance. The synergy of these improvements resulted in the model achieving the highest performance across all metrics, with no conflicting interactions among the modules, demonstrating their independence and compatibility.

Overall, the proposed detection model delivers superior performance as an assistive tool for accurate and efficient CTC detection, holding significant potential for applications in cancer early diagnosis, treatment planning, and prognosis evaluation in an end-to-end manner. Looking ahead, to meet the practical demands of laboratory settings, we will explore the feasibility of detecting CTCs using only light field images. Considering the high cost of biological antibodies and fluorescent dyes for CTC capture, aligning light field images with stained images and integrating our detection framework may offer a lower-cost alternative for automatic CTC localization and capture. This advancement could provide an efficient and cost-effective method for cancer early detection, contributing to improved global healthcare outcomes.

In medical imaging, particularly in domains like X-ray image segmentation and multi-modal imaging, the scarcity of annotated data presents a significant challenge. While basic tasks such as image classification can be easily annotated, more advanced tasks such as object detection, semantic segmentation, and multi-modal fusion require precise, labor-intensive annotations, which are often unavailable.

The proposed model addresses this issue by enabling the effective use of datasets with only basic classification annotations for more advanced tasks. This approach significantly reduces the need for extensive manual labeling, especially in resource-constrained settings. By leveraging data augmentation, the model enhances generalization across tasks like X-ray image segmentation and multi-modal imaging. It learns from sparse annotations at the classification level and extends this knowledge to tasks such as tumor segmentation in X-rays and the integration of complementary information from MRI and PET in multi-modal imaging.

In addition, the trade-offs between performance and efficiency are critical considerations when deploying advanced deep learning models, particularly in resource-constrained environments such as hospitals. Achieving an optimal balance between model performance and computational efficiency is essential. This can be accomplished through strategies such as model pruning, quantization, and the development of lightweight model variants. These techniques are designed to reduce the computational burden without significantly compromising model accuracy. Furthermore, the potential of edge computing and embedded systems should be explored, as they can enable the deployment of these models in clinical settings, ensuring that high-performance diagnostics remain accessible, even in environments with limited computational resources. In conclusion, while the proposed framework demonstrates superior accuracy, ongoing efforts to optimize model efficiency will further enhance its practical applicability, making it a valuable tool for real-world medical imaging applications.

## 6. Conclusions

In conclusion, this study presented a novel data synthesis method leveraging existing CTC classification image data to generate datasets that enable the simultaneous localization and recognition of CTCs. A hybrid dataset composed of both real and synthetic data was constructed for comprehensive evaluation of automated CTC detection tasks. Furthermore, a detection framework based on the Swin Transformer was proposed, incorporating scale and shape adapters to enhance the model’s adaptive learning capabilities with respect to the heterogeneity in CTC size and shape. The model’s performance was further optimized by designing a weighted average loss function to balance real and synthetic data, and by introducing a regularization term to promote consistent feature learning across both datasets, ultimately improving the model’s representational learning effectiveness.

In the evaluations for simultaneous CTC localization and recognition, the proposed model achieved the best performance across five metrics, with a recall of 0.9961, precision of 0.9804, specificity of 0.9975, and accuracy of 0.9960, outperforming other state-of-the-art (SOTA) models. The model also attained a mean average precision (*mAP*) of 0.9400 at IOU = 0.5, surpassing three other SOTA models (DiffusionDet, CO-DETR, and DDQ) by 3.6, 7.5, and 7.2 percentage points, respectively. These outstanding results underscore the potential of the proposed CTC detection method in applications such as cancer early diagnosis, treatment planning, and prognosis evaluation. We expect the proposed approach to integrate into clinical practices, ultimately advancing healthcare outcomes and enhancing human well-being.

## Figures and Tables

**Figure 1 sensors-24-07822-f001:**
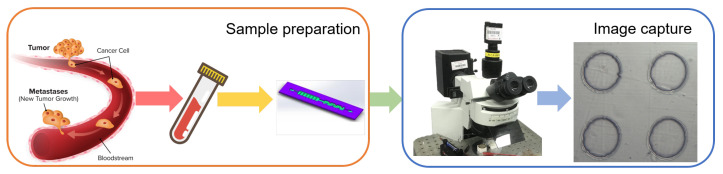
Workflow of CTC enrichment and imaging.

**Figure 2 sensors-24-07822-f002:**
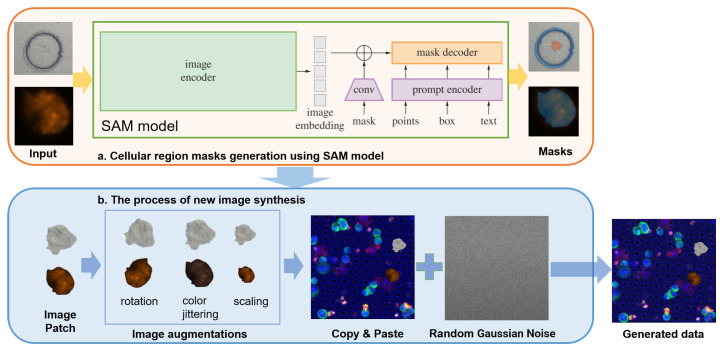
The pipeline of the image synthesis. (**a**) Cellular region masks generation using SAM model. (**b**) Image generation through patch augmentation, "copy-paste" strategy, and Gaussian noise to generate diverse training samples.

**Figure 3 sensors-24-07822-f003:**
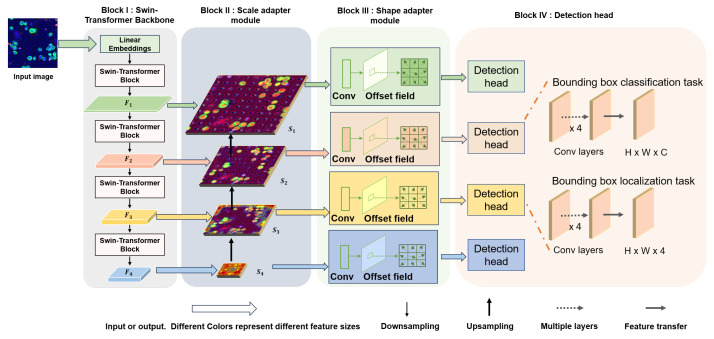
The architecture of the proposed detection framework.

**Figure 4 sensors-24-07822-f004:**
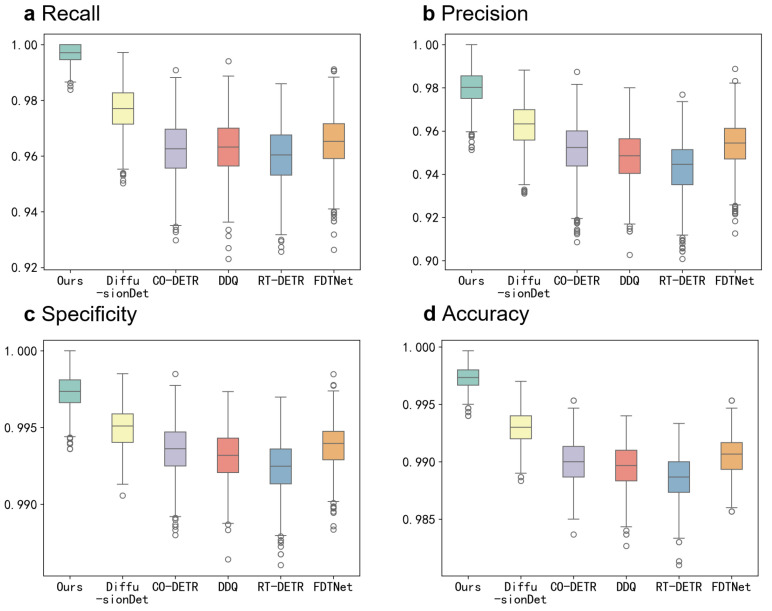
Boxplots of the recall, precision, specificity, and accuracy for the proposed model and the other five SOTA methods. (**a**) Recall for CTC detection. (**b**) Precision for CTC detection. (**c**) Specificity for CTC detection. (**d**) Accuracy for CTC detection.

**Figure 5 sensors-24-07822-f005:**
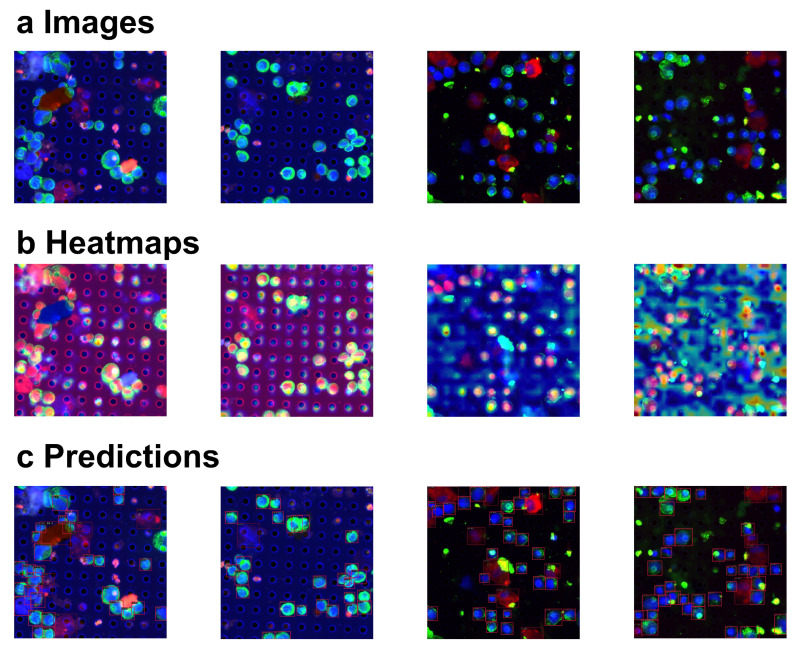
Visualizations of input images, corresponding heatmaps, and model predictions. (**a**) Input images. (**b**) Heatmaps generated for the input images. The intensity of the color, particularly the red, indicates stronger attention (**c**) Predictions made by the proposed model. The red box represents the predicted bounding box for the CTCs.

**Figure 6 sensors-24-07822-f006:**
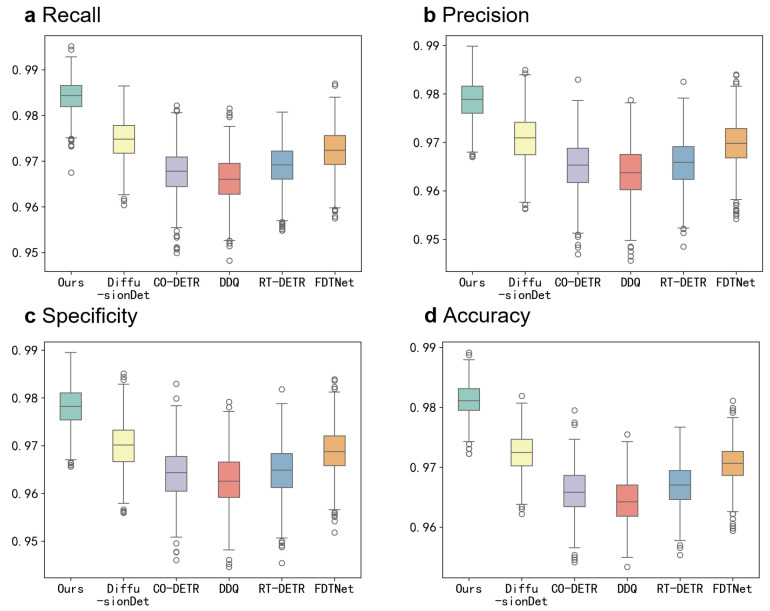
Boxplots of the recall, precision, specificity, and accuracy for the proposed model and the other five SOTA methods. (**a**) Recall for CTC detection. (**b**) Precision for CTC detection. (**c**) Specificity for CTC detection. (**d**) Accuracy for CTC detection.

**Figure 7 sensors-24-07822-f007:**
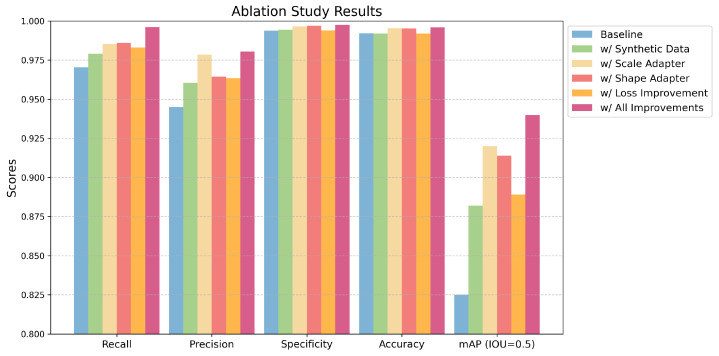
Performance evaluation of different improvements of the proposed models.

**Figure 8 sensors-24-07822-f008:**
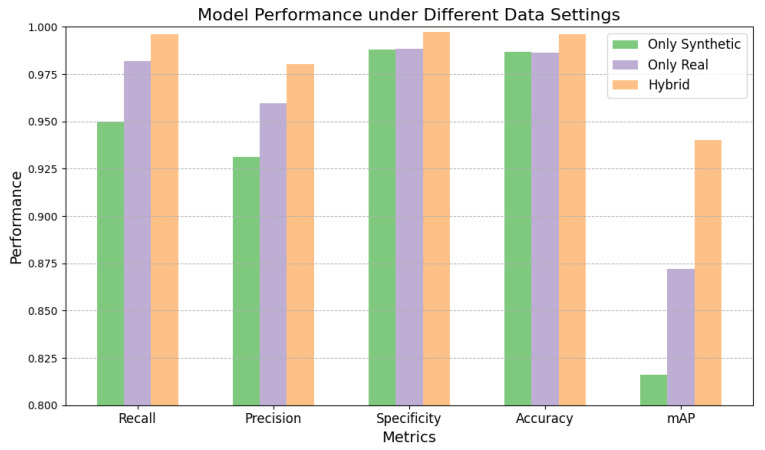
Performance evaluation of models trained on different data settings.

**Table 1 sensors-24-07822-t001:** Comparison of distributions among the constructed CTCH, public BCCD, public CTCB, and public CTCC datasets.

Dataset	Dataset Split		Localization	Classification
	**Train**	**Test**		
BCCD [37]	9957	2487	✓	✓
CTCC [10]	11,332	2834	✕	✓
CTCB [14]	1000	100	✓	✓
CTCH (proposed)	12,000	3000	✓	✓

**Table 2 sensors-24-07822-t002:** Comparative study results on the CTCH dataset.

Method	Recall	Precision	Specificity	Accuracy	*mAP* (IOU = 0.5)
ADCTC [14]	0.9600	0.9400	-	-	-
DiffusionDet [32]	0.9793	0.9635	0.9948	0.9929	0.8940
CO-DETR [33]	0.9635	0.9510	0.9931	0.9894	0.8650
DDQ [34]	0.9635	0.9540	0.9936	0.9901	0.8680
RT-DETR [35]	0.9597	0.9433	0.9925	0.9887	0.8560
FDTNet [36]	0.9654	0.9544	0.9940	0.9907	0.8685
Ours	0.9961	0.9804	0.9975	0.9960	0.9400

**Table 3 sensors-24-07822-t003:** Comparative study results on the BCCD dataset.

Method	Recall	Precision	Specificity	Accuracy	*mAP* (IOU = 0.5)
DiffusionDet [32]	0.9723	0.9708	0.9698	0.9723	0.8460
CO-DETR [33]	0.9675	0.9652	0.9641	0.9658	0.8380
DDQ [34]	0.9659	0.9636	0.9625	0.9642	0.8350
RT-DETR [35]	0.9690	0.9660	0.9650	0.9670	0.8420
FDTNet [36]	0.9722	0.9699	0.9690	0.9706	0.8450
Ours	0.9841	0.9787	0.9780	0.9811	0.8760

**Table 4 sensors-24-07822-t004:** Ablation study results.

Method	Recall	Precision	Specificity	Accuracy	*mAP* (IOU = 0.5)
Baseline	0.9704	0.9450	0.9938	0.9922	0.8250
w/ Synthetic Data	0.9791	0.9605	0.9945	0.9920	0.8820
w/ Scale Adapter	0.9854	0.9785	0.9965	0.9954	0.9200
w/ Shape Adapter	0.9860	0.9645	0.9969	0.9952	0.9140
w/ Loss Improvement	0.9830	0.9635	0.9939	0.9920	0.8890
w/ All Improvements	0.9961	0.9804	0.9975	0.9960	0.9400

## Data Availability

The dataset is available from the corresponding author on reasonable request.

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
