# Peer review of "Improving Circulating Tumor Cell Detection Using Image Synthesis and Transformer Models in Cancer Diagnostics"

_sensors, 2024, doi:10.3390/s24237822_

Round 1

Reviewer 1 Report

Comments and Suggestions for Authors

The paper proposes a novel framework for detecting circulating tumor cells (CTCs) in blood samples using synthetic image generation and a Transformer-based detection model. The Segment Anything Model (SAM) and a copy-paste strategy generate synthetic and real data to create a hybrid dataset. The model incorporates scale and shape adaptive modules to handle CTC size and shape variability. The results show significant improvements over state-of-the-art (SOTA) models, with metrics such as recall, precision, and mAP highlighting the effectiveness of the proposed approach. However, it requires some improvements mentioned below:

1.       The authors should explore additional datasets to test the model’s generalizability. For instance, using histopathology datasets like CAMELYON would demonstrate applicability to other cellular imaging tasks, while BRATS could test performance on tumor segmentation in MRI (Section 3.1).

2.       Add details on memory usage and runtime efficiency to highlight practical considerations for deploying the model in clinical settings. For example, specifying the GPU hours required for training or the latency during inference would provide valuable insights (Section 5).

3.       Briefly explain technical components like SAM, shape adaptation, and scale adaptation in simpler terms for readers unfamiliar with deep learning. For example, describing how SAM delineates boundaries without annotations or how deformable convolution adapts to shapes would improve accessibility (Section 3.3).

4.       Extend visualizations to include challenging scenarios, such as images with multiple overlapping CTCs or varying image qualities. This would illustrate the robustness of the method under diverse conditions (Section 4.2.1).

5.       Discuss potential adaptations of the method for other domains, such as X-ray image segmentation or multi-modal imaging (e.g., combining MRI and PET scans). This could inspire extensions of the methodology (Section 5).

6.       While the method shows superior accuracy, the added computational costs of synthetic data and transformer-based detection could be a limitation. Addressing trade-offs between performance and efficiency would strengthen the paper’s practical relevance (Section 5).

7.       Including newer models like nnUNet or Swin-UNet in the comparisons would validate whether the proposed enhancements offer unique advantages over well-established frameworks (Section 4.2.1).

Author Response

Dear reviewer, thank you for your time and valuable suggestions. Please see the attachment: 'point-by-point response to reviewer1's comments.pdf'

Reviewer 2 Report

Comments and Suggestions for Authors

Thanks for the possibility to review your work. This paper is about the use of SAM and Transformer for tumor cell detection. I have some revisions I would kindly ask authors to address.

1. It is recommended to introduce the data set used in the abstract.

2. The improved loss function is introduced in the abstract, but it is not mentioned in the contribution summary in Section 2. It is recommended to keep the content consistent.

3.In this paper, the segmentation model of SAM is used to delineate cell boundaries and generate masks. However, the description of this key technology of SAM is too brief, and it is recommended to introduce its structure and principle in detail in the related work section.

4. It is suggested to directly clarify the advantages of various indicators of the model compared with advanced experiments.

5. It is recommended to analyze and introduce the contents in Figure 4.

6. In section 4.2.3, it is recommended to do a summative analysis of Figure 6 rather than re-describe the data in the figure in words.

Comments on the Quality of English Language

The English could be improved to more clearly express the research.

Author Response

Dear reviewer, thank you for your time and valuable suggestions. Please see the attachment 'point-by-point response to reviewer2's comments.pdf'

Round 2

Reviewer 1 Report

Comments and Suggestions for Authors

Most of my comments are addressed. I recommend acceptance of this article. 

Reviewer 2 Report

Comments and Suggestions for Authors

All my questions have been well solved and the quality of the manuscript is also further improved. I think that it may accept for publication.